# GauDP: Reinventing Multi-Agent Collaboration through Gaussian-Image Synergy in Diffusion Policies

**Ziye Wang**[1,2†*]    **Li Kang**[3†]    **Yiran Qin**[1,4†]    **Jiahua Ma**[1]    **Zhanglin Peng**[2]
**Lei Bai**[5]    **Ruimao Zhang**[1‡]

[1]Sun Yat-sen University    [2]The University of Hong Kong    [3]Shanghai Jiao Tong University
[4]The Chinese University of Hong Kong, Shenzhen    [5]Shanghai AI Laboratory

## Abstract

Recently, effective coordination in embodied multi-agent systems remains a fundamental challenge—particularly in scenarios where agents must balance individual perspectives with global environmental awareness. Existing approaches often struggle to balance fine-grained local control with comprehensive scene understanding, resulting in limited scalability and compromised collaboration quality. In this paper, we present *GauDP*, a novel Gaussian-image synergistic representation that facilitates scalable, perception-aware imitation learning in multi-agent collaborative systems. Specifically, *GauDP* constructs a globally consistent 3D Gaussian field from decentralized RGB observations, then dynamically redistributes 3D Gaussian attributes to each agent's local perspective. This enables all agents to adaptively query task-critical features from the shared scene representation while maintaining their individual viewpoints. This design facilitates both fine-grained control and globally coherent behavior without requiring additional sensing modalities. We evaluate *GauDP* on the RoboFactory benchmark, which includes diverse multi-arm manipulation tasks. Our method achieves superior performance over existing image-based methods and approaches the effectiveness of point-cloud-driven methods, while maintaining strong scalability as the number of agents increases. Codes are available at `https://ziyeeee.github.io/gaudp.io/`.

## 1 Introduction

Multi-agent embodied collaboration [1, 2, 3, 4] is emerging as a key enabler in a wide range of real-world domains, including industrial assembly [5], surgical robotics [6], and assistive household [7] tasks. Unlike single-agent settings, multi-agent collaboration introduces a unique challenge: each agent must complete its assigned task while remaining synchronized with others to avoid catastrophic failures such as collisions or task disruptions.

Existing approaches [8, 9] for multi-agent control typically rely on two paradigms of observation. The first aggregates local observations from all agents and feeds them into a single shared policy (Fig. 1a). While local views offer fine-grained details necessary for precise manipulation, simply concatenating these observations fails to capture the joint collaborative state, often leading to misaligned execution. For instance, one arm may attempt to place food into a pot before the other has finished lifting the lid—resulting in failed coordination. The second paradigm employs a global observation of the entire environment (Fig. 1b), which provides a consistent representation for joint decision-making.

---

*Work completed by Ziye Wang as a visiting research student at Sun Yat-sen University.
†Equal contribution.
‡Corresponding author: Ruimao Zhang `ruimao.zhang@ieee.org`.

39th Conference on Neural Information Processing Systems (NeurIPS 2025).

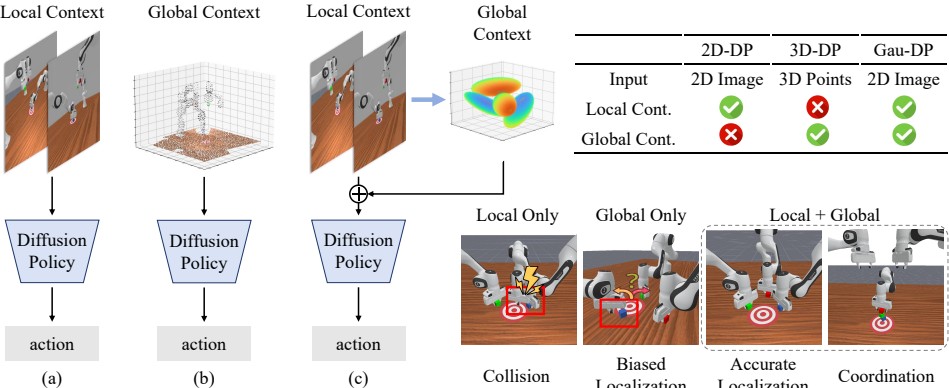

Figure 1: Both local and global context are essential in multi-agent collaboration. Comparison of multi-agent decision-making using different types of contextual information. (a) Using only local context leads to miscoordination such as collisions due to lack of global awareness. (b) Using only global context provides a holistic scene view but lacks detailed local features, resulting in inaccurate control, such as biased localization. (c) Our proposed method, *GauDP*, fuses global context, which is reconstructed from 2D local images via a shared 3D Gaussian representation, on top of local observations. This integration enables both accurate localization and coordinated execution. Our proposed method, based solely on 2D observations, effectively aggregates global context on top of the local context.

However, this approach often lacks the high-resolution, agent-specific information required for reliable low-level control, such as grasping or placement, thus reducing individual agent performance.

To address this dilemma, effectively integrating both global and local observations is crucial. However, naive fusion of these signals typically lacks 3D structural constraints, making it difficult for the model to reason about spatial relationships and agent-specific contexts. This motivates the need for a unified representation that can simultaneously encode global consistency and local precision.

To this end, we propose *GauDP*, a unified image-Gaussian representation for multi-agent embodied collaboration (Fig. 1c). Our framework first reconstructs a 3D Gaussian [10] field from the agents' local-view RGB images captured from arbitrary viewpoints. This allows the system to build a globally consistent yet spatially detailed scene representation. Each agent then dynamically queries the shared Gaussian representation to extract task-relevant features for decision-making, enabling coordination while preserving fine-grained control. Importantly, our design naturally scales to more agents without requiring architectural changes, thanks to the flexibility of the Gaussian representation.

We evaluate *GauDP* on the RoboFactory [1] benchmark across diverse multi-arm collaboration tasks. Experiments show that our method significantly outperforms image-based imitation learning methods [] and achieves performance comparable to point-cloud-based methods such as 3D Diffusion Policy [11], despite using only RGB input. Further ablation studies demonstrate *GauDP*'s robustness and scalability as the number of agents increases. Visualization results confirm its ability to integrate multi-agent observations into a high-quality 3D global representation that improves decision accuracy.

Our contributions are summarized as follows: (1) We introduce *GauDP*, a unified framework that integrates local and global observations via 3D Gaussian fields for multi-agent embodied collaboration. (2) We design a dynamic representation selection mechanism that enables each agent to reason over shared 3D context while maintaining individual precision. (3) We demonstrate the effectiveness and scalability of *GauDP* on the RoboFactory benchmark, achieving strong performance with only RGB input.

## 2 Related Work

### 2.1 3D Reconstruction from Multi-View Images

The advent of Neural Radiance Fields (NeRF) [12] and 3D Gaussian Splatting (3DGS) [13] has significantly advanced 3D scene reconstruction by representing entire environments as a unified set

of spatially distributed primitives. These methods are capable of not only accurately reconstructing photorealistic visual appearances but also capturing the underlying 3D geometric structure of a scene from multi-view images. However, achieving high-fidelity reconstructions typically requires densely sampled input views and lengthy optimization time for each scene.

To address this, a growing body of work has focused on adapting NeRF [14, 15, 16, 17, 18] and 3DGS [10, 19, 20, 21, 22, 23] to operate under sparse-view conditions. These approaches typically introduce additional priors, such as semantic information or geometric constraints, to regularize the inherently ill-posed problem of reconstruction from limited viewpoints. Besides, they still often rely on accurate camera poses and sufficient overlap among the views, which are difficult to obtain in real-world robotic manipulation scenarios.

Beyond reconstructing high-fidelity 3D scenes, traditional Structure-from-Motion (SfM) pipelines [24] estimate both 3D structure and camera poses based on sparse feature correspondences. While SfM remains effective in many cases, its performance significantly degrades under wide baselines or extremely sparse views, where reliable feature matching becomes challenging. Recently, learning-based approaches have emerged that directly infer dense 3D geometry from a small number of images [25, 26, 27, 28]. These methods mark a shift toward end-to-end systems that implicitly learn geometric relationships, enabling 3D structure estimation even from as few as two input views.

## 2.2 Robot Manipulation

Behavioral Cloning (BC)[29, 30, 31, 32, 33] trains policies using pre-recorded human demonstrations to directly imitate expert behaviors, whereas Offline Reinforcement Learning (ORL)[34, 35, 36] refines action selection through reward maximization over large-scale fixed datasets. While BC directly mimics demonstrated behavior, ORL enables further policy improvement by optimizing over offline rewards. Generative approaches have expanded the landscape of policy learning: Action Chunking with Transformers (ACT) combines Transformer architectures with conditional variational autoencoders to capture temporal dependencies in sequential decision-making [37, 38, 39]. More recently, diffusion-based frameworks have shown strong potential in robotic imitation learning due to their high-fidelity trajectory generation. Notable examples include Diffusion Policy [40] and its 3D extension [11], which leverages point cloud inputs to enhance spatial reasoning. The stochastic generative nature of these models makes them especially effective in capturing multimodal action distributions. Demonstration acquisition primarily relies on human-operated robotic systems across diverse tasks [41, 32, 42, 30], while simulator-based trajectory synthesis has emerged as a scalable alternative [43, 44, 45, 46, 47]. Simulated environments allow for controlled task variation and embodiment flexibility. However, existing systems largely focus on single-agent scenarios. Effective data-driven policy learning for multi-agent robotic manipulation—particularly in settings involving coordination among multi-agent—remain significantly underexplored.

## 3 Method

### 3.1 Preliminary

**3D Gaussian Splatting (3DGS).** 3D Gaussian Splatting [13] represents a 3D scene as a collection of spatially distributed anisotropic Gaussian primitives. Each Gaussian is parameterized by a 3D mean position $\boldsymbol{\mu} \in \mathbb{R}^3$, a scaling vector $\boldsymbol{s} \in \mathbb{R}^3$, a rotation represented by a unit quaternion $\boldsymbol{r} \in \mathbb{R}^4$, an opacity value $\alpha \in \mathbb{R}$, and color information $\boldsymbol{c} \in \mathbb{R}^3$. To model view-dependent effects, the RGB color can be modulated by additional spherical harmonics coefficients $\boldsymbol{h} \in \mathbb{R}^k$. Therefore, a group of Gaussians can be represented as $\mathcal{G} = \{\cup(\boldsymbol{\mu}_i, \boldsymbol{s}_i, \boldsymbol{r}_i, \alpha_i, \boldsymbol{c}_i, \boldsymbol{h}_i)\}$.

Rendering in 3DGS is performed through a differentiable rasterization process that computes each Gaussian's contribution to the image plane. First, all Gaussians are projected into the camera coordinate system. The screen space is then divided into tiles, and Gaussians falling outside the view frustum are efficiently culled to reduce computation. Finally, for each pixel, visible Gaussians are sorted in view-space depth order, and their contributions are composited via alpha blending.

This rendering pipeline enables accurate and efficient supervision of the underlying 3D structure. During optimization, the parameters of the Gaussians are updated to minimize the discrepancy between the rendered images and the ground-truth observations across multiple camera views.

Importantly, **when the optimized Gaussian representation yields rendered images that are consistent with the ground-truth across views, it can be regarded as a faithful reconstruction of the true scene geometry**. Formally, let $\mathcal{G}_A$ and $\mathcal{G}_B$ denote two distinct 3D Gaussian configurations, and let $\mathcal{R}(\mathcal{G}, v)$ denote the rendered image of $\mathcal{G}$ under viewpoint $v$. If the renderings of $\mathcal{G}_A$ and $\mathcal{G}_B$ are identical across all training views $\mathcal{V} = v_1, v_2, \ldots, v_N$:

$$\forall v \in \mathcal{V}, \quad \mathcal{R}(\mathcal{G}_A, v) = \mathcal{R}(\mathcal{G}_B, v),$$

then $\mathcal{G}_A$ and $\mathcal{G}_B$ must be geometrically equivalent, i.e., $\mathcal{G}_A \equiv \mathcal{G}_B$. Conversely, if they are not geometrically equivalent, there must exist at least one view $v \in \mathcal{V}$ such that their renderings differ:

$$\mathcal{G}_A \not\equiv \mathcal{G}_B \quad \Rightarrow \quad \exists v \in \mathcal{V}, \quad \mathcal{R}(\mathcal{G}_A, v) \neq \mathcal{R}(\mathcal{G}_B, v).$$

This implies that multi-view consistency effectively constrains the optimization to a unique, geometrically faithful solution in a self-supervised manner.

**Diffusion Policy (DP).** Diffusion Policy [48] formulates action generation as a conditional denoising diffusion process. Given a sequence of past observations $\mathcal{O} = \{\mathcal{I}_1, \mathcal{I}_2, \ldots, \mathcal{I}_N\}$, the goal is to generate a future action sequence $\boldsymbol{a} = \{a_1, a_2, \ldots, a_L\}$.

The target action sequence $\boldsymbol{a}$ is gradually perturbed by Gaussian noise through a forward diffusion process:

$$q(\boldsymbol{a}^k \mid \boldsymbol{a}^{k-1}) = \mathcal{N}\left(\sqrt{1 - \beta_k}\, \boldsymbol{a}^{k-1},\, \beta_k \mathbf{I}\right), \quad k = 1, \ldots, K,$$

where $\beta_k$ is the noise variance schedule. The reverse process learns to iteratively denoise a random sample $\boldsymbol{a}^K \sim \mathcal{N}(\mathbf{0}, \mathbf{I})$ back to a clean action sequence, conditioned on observations $\mathcal{O}$:

$$p_\Phi(\boldsymbol{a}^{k-1} \mid \boldsymbol{a}^k, \mathcal{O}) = \mathcal{N}\left(\boldsymbol{\mu}_\Phi(\boldsymbol{a}^k, \mathcal{O}, k), \Sigma_\Phi(\boldsymbol{a}^k, \mathcal{O}, k)\right),$$

where $\Phi$ denotes the parameters of the conditional denoising network. Through iterative denoising, the learned policy $\pi_\Phi$ generates action trajectories by:

$$\pi_\Phi(\boldsymbol{a} \mid \mathcal{O}) = \mathbb{E}_{\boldsymbol{a}^K \sim \mathcal{N}(\mathbf{0}, \mathbf{I})} \left[ \prod_{k=1}^{K} p_\Phi(\boldsymbol{a}^{K-k} \mid \boldsymbol{a}^{K-k+1}, \mathcal{O}) \right].$$

### 3.2 Problem Formulation

We consider the problem of predicting future action sequences for multi-arm embodied agents based on multi-view visual observations. Let $\mathcal{O} = \{\mathcal{I}_1, \mathcal{I}_2, \ldots, \mathcal{I}_N\}$ denote a set of synchronized observations captured from $N$ views. Each image $\mathcal{I}_i$ captures the scene from a unique perspective, offering complementary geometric information. The prediction target is a sequence of future actions $\boldsymbol{a} = \{a_1, a_2, \ldots, a_L\}$, where $a_t \in \mathbb{R}^d$ represents the control signal at timestep $t$.

Rather than directly predicting $\boldsymbol{a}$ from 2D image features, we first reconstruct a compact and differentiable 3D Gaussian representation $\mathcal{G}$ as a global context from $\mathcal{O}$. We define a conditional policy $\pi_\Phi$ parameterized by $\Phi$ that generates the action sequence $\boldsymbol{a}$ conditioned on both the raw visual inputs $\mathcal{O}$ and the reconstructed 3D representation $\mathcal{G}$:

$$\pi_\Phi(\boldsymbol{a} \mid \mathcal{O}) := \pi_\Phi(\boldsymbol{a} \mid \mathcal{O}, \mathcal{G}).$$

Where, $\mathcal{G} = \mathcal{F}(\mathcal{O})$ and $\mathcal{F}(\cdot)$ is a mapping from multi-view observations to a set of Gaussians. To model the complex conditional distribution over action sequences, we adopt the Diffusion Policy framework. Let $\boldsymbol{a}$ denote the target action sequence, and let $\boldsymbol{a}^K \sim \mathcal{N}(\mathbf{0}, \mathbf{I})$ be a sample from an isotropic Gaussian prior. The generative process denoises $\boldsymbol{a}^K$ through a learned reverse process conditioned on $(\mathcal{O}, \mathcal{G})$:

$$\pi_\Phi(\boldsymbol{a}_t \mid \mathcal{O}, \mathcal{G}) = \mathbb{E}_{\boldsymbol{a}_t^K \sim \mathcal{N}(0, \mathbf{I})} \left[ \prod_{i=1}^{K} p_\Phi(\boldsymbol{a}_t^{K-i} \mid \boldsymbol{a}_t^{K-i+1}, \mathcal{O}, \mathcal{G}) \right],$$

where $p_\Phi$ denotes the learned denoising transition at each timestep. This framework allows the policy to generate realistic and context-aware action trajectories without relying on strong priors over the action space.

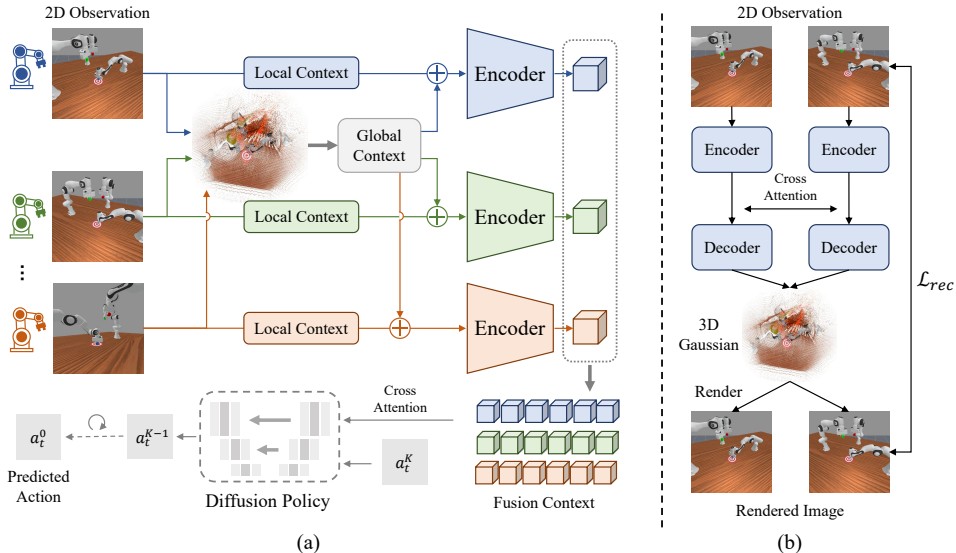

Figure 2: **(a)** Overview of the proposed *GauDP* framework for multi-agent imitation learning. Each agent extracts a local context from its 2D observation. A shared 3D Gaussian field is constructed from all views to form the global context, which is fused with the local context and passed through an encoder. The resulting per-agent features are processed by a diffusion policy via cross-attention to predict actions. **(b)** Pipeline for constructing the global Gaussian field. Multi-view images are encoded and aggregated via cross-attention, followed by a reconstruction loss $\mathcal{L}_{\text{rec}}$ between rendered and input views to ensure consistency.

## 3.3 Overview

Mainstream approaches typically train policies to predict future actions directly from either image observations or global point clouds. However, these methods face limitations in multi-agent collaborative tasks: they either rely solely on local observations for each agent or require access to a centralized global point cloud, which restricts both accurate localization and effective coordination among agents. To address these challenges, we propose a novel framework solely based on image observations that enables collaborative multi-agent manipulation through effective global context integration. Specifically, our method fuses multi-view observations to reconstruct a unified global context and selectively distributes task-relevant features back to individual agents as needed. This design not only enhances inter-agent coordination but also improves precise perception and localization. An overview of the proposed framework is illustrated in Fig. 2(a).

## 3.4 Global Context Reconstruction

In this work, we define a global context as a unified, view-independent representation built within a common 3D coordinate space. This representation should not only preserve color information from raw multi-view image observations, but also restore the underlying 3D structure of the scene reconstructed from these views. To achieve this, we design a framework that reconstructs 3D scenes in a self-supervised manner using only the multi-view 2D observations typically employed for training diffusion policies. Our reconstruction framework is built upon 3D Gaussian Splatting (3DGS). However, conventional 3DGS methods suffer from two major limitations: First, they require densely sampled views with accurate camera poses. Second, they demand scene-specific optimization that can take several minutes per scene. These constraints render them impractical for embodied scenarios, where rapid adaptation and generalization are essential.

To overcome these challenges, we adopt Noposplat [49], a feed-forward network capable of directly reconstructing 3D Gaussian representations from sparse and unposed views. We further fine-tune the pretrained Noposplat model using multi-view observations collected from our robotic manipulation scenarios, which are the same data used to train our downstream diffusion policy.

As illustrated in Fig.2(b), each RGB image is independently encoded by a shared-weight ViT [50] encoder across all views. The resulting per-view features are then passed through a cross-view ViT decoder, which fuses information across different perspectives using cross-attention layers in each transformer block. Finally, a Gaussian parameter prediction head estimates a set of 3D Gaussians for each pixel based on the fused features. This process can be expressed as:

$$\mathcal{G}_i = \mathcal{F}(\mathbf{x}_i), \quad \forall i \in \mathcal{I},$$

where $\mathbf{x}_i$ denotes the fused feature at pixel $i$, $\mathcal{F}(\cdot)$ is the mapping network, and $\mathcal{G}_i \in \mathbb{R}^{C_{\mathcal{G}} \times H \times W}$ represents the estimated parameters of the corresponding 3D Gaussian.

To further improve the fidelity of the reconstructed 3D structure, we introduce an additional depth supervision during fine-tuning. Specifically, in the rendering process, each estimated 3D Gaussian is projected onto the camera coordinate system. Instead of computing the RGB contribution of each Gaussian to the image pixels, we compute the contribution of its projected depth to the corresponding pixel. This depth rendering process yields a synthetic depth map $\hat{D}$, which can then be supervised using available ground-truth depth $D$ via a reconstruction loss $\mathcal{L}_{\text{depth}}$. This depth-based supervision provides stronger geometric guidance and encourages the model to recover 3D Gaussians that are more consistent with the actual scene geometry. The overall reconstruction loss is defined as:

$$\mathcal{L}_{\text{rec}} = \mathcal{L}_{\text{rgb}} + \alpha \cdot \mathcal{L}_{\text{depth}},$$

where $\alpha$ is a balancing weight that controls the influence of the depth supervision.

It is worth noting that depth maps and camera poses are used only during the fine-tuning stage of Noposplat. During policy training and inference, our framework solely relies on multi-view RGB observations to infer the 3D Gaussians, making it lightweight and pose-free at deployment time.

## 3.5 Global Context Allocation and Pixel-level Synergy

The reconstructed global context encodes rich multi-view and multi-agent information, capturing both the semantic and geometric structure of the scene. However, directly feeding the entire global context to each agent is suboptimal, as it introduces irrelevant information and may interfere with the agent's ability to focus on task-relevant cues from its own perspective. Moreover, effective synergy between global and local context remains underexplored. Existing approaches typically aggregate global and local information only at a coarse level, which fails to capture fine-grained spatial alignment and task-specific dependencies. This coarse fusion strategy may lead to diluted feature representations and impaired action reasoning, especially in densely interactive multi-agent scenarios.

To address these challenges, we introduce a selective global context dispatch mechanism along with a pixel-aligned fusion strategy for fine-grained integration of global and local information. Recall that in Section 3.4, we reconstruct 3D Gaussians by aggregating multi-view image tokens via cross-attention. Each predicted Gaussian encodes both visual appearance and geometry within a unified global coordinate system, while remaining naturally aligned with the input image pixels from which it was derived. Leveraging this alignment, we selectively dispatch the predicted 3D Gaussians back to the corresponding agent's observation frame based on their image of origin. Instead of distributing the entire global context indiscriminately, each agent receives only the subset of Gaussians associated with its own view. These Gaussians have already integrated information from other views during reconstruction, thus providing a distilled and relevant global summary for that agent.

For synergistic fusion, we transform the selected Gaussians back into a 2D grid that matches the spatial dimensions of the original image. These global context features are then concatenated with the agent's local image features and passed through a lightweight convolutional fusion module, which learns to combine the complementary strengths of local perception and global understanding.

This design ensures that each agent benefits from a targeted and contextually relevant global representation, while preserving spatial consistency and enabling pixel-level synergy between local and global cues, both of which are critical for precise and coordinated action planning in multi-agent manipulation tasks.

Table 1: Quantitative Comparison of 3D Gaussian Reconstruction. Improved visual quality reflects higher accuracy in 3D reconstruction.

| Method | PSNR ↑ | SSIM ↑ | LPIPS ↓ |
|--------|--------|--------|---------|
| Pretrain | 17.918 | 0.580 | 0.492 |
| **Ours** | **23.424** | **0.779** | **0.148** |

## 4 Experiment

### 4.1 Experiment Setup

**Dataset.** Imitation learning in multi-agent collaborative manipulation presents significant challenges due to the high levels of complexity, coordination, synchronization, and symmetry awareness required, making it difficult to collect high-quality data in real-world scenarios. To address this issue, we leverage the RoboFactory benchmark [1], an automated data collection framework specifically designed for embodied multi-agent systems. We select 6 tasks from RoboFactory involving collaborative manipulation using two to four robotic arms. These tasks are designed to cover a range of coordination complexities and physical interaction patterns. Please refer to the Appendix for detailed descriptions of each task.

**Baseline.** Existing diffusion-policy-based approaches predominantly focus on either 2D or 3D modalities. To ensure a comprehensive and fair comparison, we evaluate our method against several representative baselines in both domains. For 2D vision-based observations, we adopt Diffusion Policy [48] and 2D Dense Policy [51]; for 3D input modalities, we include 3D Diffusion Policy [11] and 3D Dense Policy [51] as baselines. For fair comparison, we maintain a consistent visual backbone across all methods: ResNet-18 is used for 2D visual inputs following the original Diffusion Policy, and a lightweight MLP is used for 3D data as in 3D Diffusion Policy.

**Experiment setting.** We use success rate as the primary evaluation metric to assess the effectiveness of each policy. Evaluation is performed every 100 training epochs over 100 episodes per policy. All experiments are implemented using the PyTorch framework and conducted on a single NVIDIA A800 GPU. Policies are trained for 100 epochs using a batch size of 32. We adopt the Adam optimizer with an initial learning rate of $10^{-4}$, combined with a warm-up phase followed by cosine decay scheduling. To ensure a fair comparison, all baseline and ablation models are trained using the same set of hyperparameters and optimization settings.

For fair comparison, our proposed *GauDP* and all baseline methods are trained under identical hyperparameters and optimization settings following standard Diffusion Policy benchmarks. Specifically, we use an action prediction horizon of 8, 3 observation steps, and 6 action execution steps. Both *GauDP* and DP adopt DDPM with 100 denoising steps, while DP3 employs DDIM with the same number of steps.

### 4.2 Experiment Results

**Reconstruction Results.** As discussed in Section 3.1, higher-quality rendered images indicate more accurate reconstruction of the underlying 3D scene, including both geometry and color information. To evaluate reconstruction performance, we conduct experiments where the full scene is reconstructed using observations from only two reference viewpoints. Quantitative results are summarized in Table 1, and qualitative comparisons of the rendered Gaussians from both reference and novel views are shown in Figure 3.

Our finetuned model significantly outperforms the pretrained baseline, producing reconstructions that are not only sharper and more detailed, but also more faithful to the original scene geometry and appearance. As shown in Figure 3, our method yields consistent and high-fidelity renderings across both reference and novel views, with clearly defined object boundaries. In contrast, the pretrained model often produces blurry and distorted results, with noticeable discrepancies between the reconstructions and the original images. Besides, in the first two columns, the reconstructed positions of the robot arms differ considerably from those in the ground truth. Our method consistently

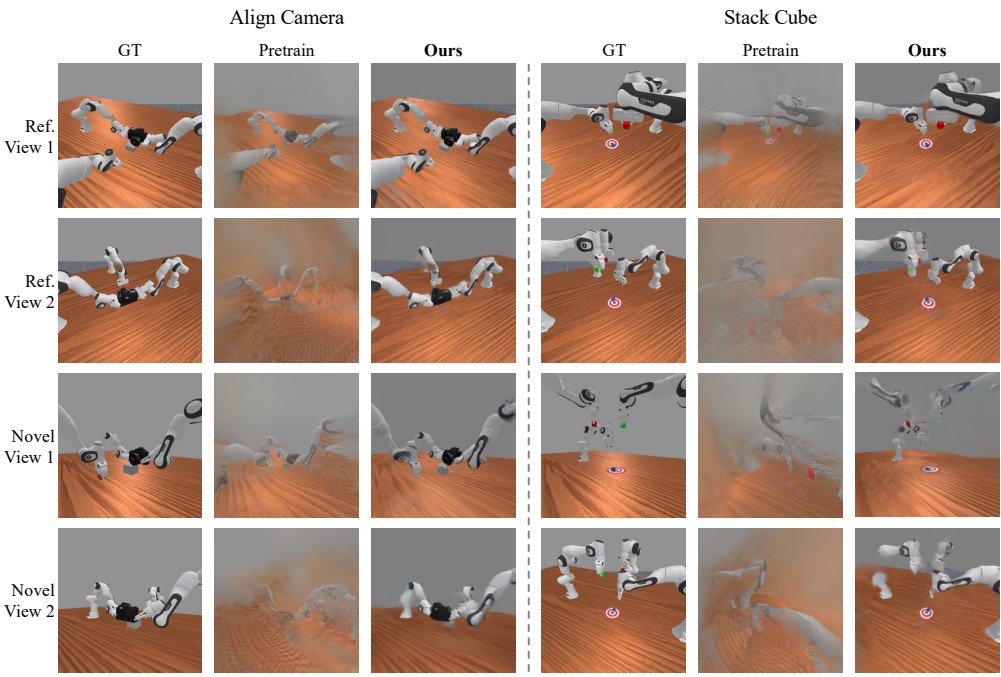

Figure 3: Visualization of Reconstruction Results. Our method achieves significantly improved reconstruction quality.

Table 2: Success rate comparison across multi-agent manipulation tasks with different 2–4 arms setting. Underlines denote the best baseline of point cloud-based diffusion policy(DP); **bold** highlights the best of image-based DP. *GauDP* achieves the highest average performance across all settings.

| Method | 2 Arms | | | 3 Arms | | 4 Arms | Avg |
|---|---|---|---|---|---|---|---|
| | Lift Barrier | Place Food | Stack Cube | Align Camera | Stack Cube | Take Photo | |
| DP3(XYZ) [11] | 30% | 21% | 1% | 3% | 0% | 9% | 10.67% |
| DP3(XYZ+RGB) [11] | 31% | 25% | 1% | 18% | 0% | 11% | 14.33% |
| 3D Dense Policy [51] | 28% | 18% | 0% | 0% | 0% | 7% | 8.83% |
| DP [48] | 9% | 12% | **6%** | 3% | 0% | 0% | 5.00% |
| 2D Dense Policy [51] | 3% | 2% | 0% | 0% | 0% | **9%** | 2.33% |
| *GauDP* | **72%** | **15%** | 2% | **26%** | 0% | 3% | **19.67%** |

maintains high structural fidelity and visual consistency, demonstrating its effectiveness in capturing the accurate 3D structure of the scene.

**Point Cloud-based Diffusion Policy.** The DP3 model leverages 3D point cloud observations to inform multi-agent manipulation. As shown in the table 2, this policy achieves moderate performance across two-arm tasks, such as 30% on Lift Barrier and 21% on Place Food, indicating that point cloud inputs capture sufficient geometric structure for spatially grounded actions. However, its effectiveness drops sharply in tasks with more agents and higher coordination requirements—such as only 1% on Stack Cubes (2 arms). These results suggest that point cloud-based methods lack the fine-grained local control afforded by geometric inputs like point clouds, which limits their effectiveness in precision-critical tasks such as stacking cubes, especially in multi-arm settings.y in tasks with more agents and higher coordination requirements.

**Image-based Diffusion Policy.** As shown in the table 2, our method GauDP-prefuse significantly outperforms prior 2D diffusion policies across multiple tasks. Compared to DP[40] and 2D Dense

Table 3: Training and inference efficiency on the *Lift Barrier* task (Training on A100 GPU; inference on NVIDIA RTX 5090 GPU).

| Method | Training Time (GPU h) | Inference Speed (FPS) |
|--------|-----------------------|-----------------------|
| DP | 4.8 | 1.49 |
| DP3 | 2.5 | 1.57 |
| *GauDP* | 6.5 | 1.28 |

Table 4: Real-robot performance comparison across three multi-agent collaboration tasks. Each score in the table is reported as $m/n$, where $m$ denotes the number of successful executions and $n$ represents the total number of rollouts performed for that task.

| Method | Card Box Stacking | | | Card Box Handover | | Grab Roller | |
|--------|-------------------|-------------|---------|-------------------|----------------|-------------|---------|
| | Place Succ. | Stack Succ. | Succ. | Place Succ. | Handover Succ. | Succ. | Succ. |
| DP | 19/30 | 11/19 | 11/30 | 22/30 | 14/22 | 14/30 | 22/30 |
| *GauDP* | **23/30** | **17/23** | **17/30** | **24/30** | **19/24** | **19/30** | **27/30** |

Policy[51], which struggle across the board (mostly below 10%), *GauDP* achieves a remarkable 72% success rate in the Lift Barrier task, indicating its superior ability to model geometry-aware visual representations. Additionally, *GauDP* shows strong performance in tasks requiring semantic alignment, such as Align Camera (26%), where other methods fail to generalize. These results demonstrate that integrating geometric priors into image-based pipelines enables better spatial understanding and more effective multi-agent coordination, especially in tasks with complex embodiment and scene variability.

**Training and Inference Efficiency.** We evaluate the training and inference efficiency of *GauDP* compared with diffusion-based baselines. As shown in Table 3, GauDP requires slightly longer training time due to its geometry-aware modules, but maintains comparable inference speed while offering superior performance.

**Real-World Experiments.** To further verify the practicality, we conduct real-robot experiments on three representative multi-agent collaboration tasks: *Card Box Stacking*, *Card Box Handover*, and *Grab Roller*. As shown in Table 4, *GauDP* consistently outperforms DP across all tasks, particularly in stacking and handover scenarios that require precise spatial coordination.

**Discussion.** As shown in Table 2, our method ***GauDP*** achieves the highest average success rate of 19.67% across diverse multi-agent manipulation tasks, significantly outperforming all baselines, including those relying on 3D point cloud inputs such as DP3 [11] and 3D Dense Policy [51]. Notably, while our method operates solely on 2D RGB inputs, it surpasses several 3D-based counterparts in tasks that require global coordination (e.g., Align Camera: 26% vs. 18%) and even matches them in fine-grained manipulation tasks (e.g., Stack Cube: 2% vs. 1%). These results highlight that our approach's strength lies not in the modality itself, but in the design of visual representations that fuse both local visual details and global spatial context. This balance enables agents to perceive geometric structure from images and reason over scene-level relationships, allowing for generalizable cooperation without relying on explicit 3D geometry.

## 4.3 Ablation Study

**Ablation on the Coordinate System of Gaussians.** We replace the original camera-coordinate parameterization of Gaussians with a unified world-coordinate system aligned to the first observation frame. Results show that using local coordinates performs better, as it preserves agent-centric spatial relationships and avoids alignment errors across diverse viewpoints. **Ablation on Fusion Strategies for Local and Global Context.** We replace the default fine-grained, pixel-level fusion with a coarse feature-level concatenation of independently encoded modalities. Performance declines with this coarse fusion, likely due to the loss of spatial alignment and fine-grained cross-modal reasoning. **Ablation on the Role of Image and Gaussian.** We remove either the image input or the Gaussian

Table 5: Ablation study on key components of **GauDP** across multi-agent manipulation tasks. **Bold** highlights the best among image-based configurations. Our full model achieves the highest average success rate, demonstrating the effectiveness of combining geometric and visual cues.

| Method | 2 Arms | | | 3 Arms | | 4 Arms | Avg |
| --- | --- | --- | --- | --- | --- | --- | --- |
| | Lift Barrier | Place Food | Stack Cube | Align Camera | Stack Cube | Take Photo | |
| w/ unify coor. | 30% | 1% | **8%** | 26% | 0% | 0% | 10.83% |
| w/o prefuse | 2% | 4% | 0% | 1% | 0% | 0% | 1.17% |
| w/o Image | 32% | 7% | 0% | **28%** | 0% | 0% | 11.17% |
| w/o Gaussian | 9% | 12% | 6% | 3% | 0% | 0% | 5.00% |
| **Ours** | **72%** | **15%** | 2% | 26% | 0% | **3%** | **19.67%** |

representation during policy training and inference. Using both inputs achieves the best results, as images provide appearance cues while Gaussians supply global geometric context.

## 5 Conclusion

In this paper, we present *GauDP* a novel framework for multi-agent collaboration through Gaussian-image synergy in diffusion policies. Specifically, *GauDP* a globally consistent 3D Gaussian representation from the local RGB observations of each agent and reallocates the Gaussian information back to individual agents. This process significantly enhances each agent's perception of the global task information, thereby boosting the success rate of complex collaborative tasks. We evaluate the performance of *GauDP* using the RoboFactory benchmark, which features a diverse set of multi-arm manipulation tasks. As the number of agents increases, *GauDP* not only outperforms existing image-based methods but also matches the effectiveness of point-cloud-driven methods. Future directions will focus on: (1) designing Gaussian representations that are more suitable as inputs for the Vision-Language-Action model to enhance its capabilities in multi-agent collaboration; and (2) leveraging Gaussians to improve the representation of dynamic scenes, enabling them to play a role in world models designed for multi-agent environments.

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
