# OpenReview forum: "GauDP: Reinventing Multi-Agent Collaboration through Gaussian-Image Synergy in Diffusion Policies"
_NeurIPS.cc/2025/Conference — NeurIPS 2025 poster_

### Official Review · Reviewer_CMJx · 2025-06-27

**Clarity:** 3
**Significance:** 3
**Originality:** 3
**Rating:** 5
**Confidence:** 3

**Summary:**

This paper tackles the multi-agent collaboration challenge for robotic manipulation tasks. It proposes a framework, named GauDP, that integrates local and global observations from different agents via 3D Gaussian fields so that each agent can reason over its own perception results enhanced by the shared 3D context. GauDP is shown to outperform existing work (other diffusion policy based frameworks) that relies only on 2d or 3d observations (but not both) on the RoboFactory benchmark.

**Questions:**

See "weaknesses"

**Ethical Concerns:**

["NO or VERY MINOR ethics concerns only"]

**Final Justification:**

I maintain my original rating.

**Limitations:**

yes

**Quality:**

3

**Strengths And Weaknesses:**

Strengths:

+ The work is well-motivated as the shared global and local observation helps collaboration between agents for robotic tasks.
+ The algorithmic design is clean and reasonable with a good amount of ablation studies (and experimental comparisons) in a relatively fair setup.
+ The results seem convincing to me.

Weaknesses:

+ The proposed approach relies on noposplat. What if the generated 3DGS results are not ideal? How much does it matter? In case it is relatively robust, does it mean there might be better "latent" representation of the shared 3D scene representation (say, affordance-aware and not an explicit repr) that benefits downstream robotic controls.

---

> ### Author Rebuttal · Authors · 2025-07-31
>
> **[Q1] What if the generated 3DGS results are not ideal?**
>
> **[A1]** Our 3D Gaussian estimation module has been pretrained and finetuned on large-scale datasets that encompass a wide range of variations, including diverse viewpoints, lighting conditions, and occlusions. To further assess its robustness, we conducted additional experiments that introduce challenging lighting variations—including random changes in light color, direction, and intensity—as well as random distractor objects (10 task-irrelevant objects randomly placed beside the manipulated objects on the table), following RoboTwin 2.0.. Despite these perturbations, our method maintained SOTA performance, achieving the highest task success rate:
>
> |             | DP  | DP3 | GauDP |
> |-------------|-----|-----|-------|
> | Grab Roller | 20% | 46% | 50%   |
>
> This demonstrates that our Gaussian estimation is robust and reliable even in complex environments.
>
> **[Q2] How much do the 3DGS results matter?**
>
> **[A2]** The 3D Gaussian representations serve two critical roles in our framework. First, they act as a shared global scene representation that facilitates coordination among multiple manipulators in collaborative settings. Second, they serve as a self-supervised structural prior that helps each local agent infer fine-grained spatial cues from 2D observations. In both respects, the quality of the 3DGS representation significantly contributes to the effectiveness of our manipulation pipeline.
>
> **[Q3] Does it mean there might be a better "latent" representation of the shared 3D scene representation?**
>
> **[A3]** Yes! We appreciate the reviewer for raising this insightful point. This line of thought aligns well with our ongoing research. The current work represents our initial attempt to incorporate 3D Gaussian representations into manipulation tasks, where we explicitly use Gaussians to represent low-level geometry and appearance. However, we believe there is considerable potential in developing a more expressive and compact latent representation.
>
> In particular, we are currently exploring a Gaussian tokenization framework that integrates high-level semantic cues and latent action information into the 3D representation. This would enable the representation to not only reflect the spatial structure of the scene but also better support reasoning and action generation. We envision such a latent representation as a more efficient interface for downstream tasks, including integration with VLMs or VLAs.

---

> > ### Comment · Area_Chair_Hrqd · 2025-08-04
> >
> > Dear Reviewer,
> >
> > Please respond to the rebuttal. Thanks.
> >
> > AC.

---

> > > ### Comment · Area_Chair_Hrqd · 2025-08-05
> > >
> > > Dear Reviewer,
> > >
> > > Please comment on the rebuttal. Thanks.
> > >
> > > AC.

---

> > ### Comment · Reviewer_CMJx · 2025-08-07
> >
> > Thanks for the authors' responses. My concerns have been solved.

---

### Official Review · Reviewer_o7bR · 2025-06-30

**Clarity:** 4
**Significance:** 3
**Originality:** 3
**Rating:** 4
**Confidence:** 4

**Summary:**

This paper presents a novel way of encoding multi-agent spatial information with 3D Gaussian representations that fuses both local observation and global context. It achieves superior results compared to other image-based representations on selective tasks, while comparable to point cloud 3D representations. The method is especially effective in tasks that require visual alignment rather than fixed coordinate alignments.

**Questions:**

1. How is your method's per-step latency compared to other 2D/3D based methods? Though GauDP is performed with 2D inputs, it is possible that 3D Gaussian reconstruction in each environment step will be significantly more expensive than even 3D point cloud based methods.
2. Is your method robust against real world disturbance in 2D observations like different lighting conditions / errors in execution?
3. What is the core difference in representation that results in the significant success rate increase of Lift Barrier, even comparing to 3D diffusion policy?

I will consider increasing my score if the author provide additional details addressing my questions above.

**Ethical Concerns:**

["NO or VERY MINOR ethics concerns only"]

**Final Justification:**

The authors addressed my concerns properly. Admittedly, the method achieves higher success rates at the cost of a 2D Gaussian overhead. This is acceptable but not enough to qualify for a score of 5 in my opinion. Therefore, my score stays 4.

**Limitations:**

yes

**Quality:**

3

**Strengths And Weaknesses:**

Strengths:
1. It is a novel approach to use multi-view 2D images as 3D prior in embodied cooperation.
2. The method performs well in Lift-barrier and Align-camera tasks, both considered difficult for image-based diffusion policies.

Weaknesses:
1. The use of 3D reconstruction potentially makes latency high in completing tasks.
2. Experiments done only in simulated environments where 3D reconstruction works really well by fine-tuning.

---

> ### Author Rebuttal · Authors · 2025-07-31
>
> We sincerely thank the reviewer for the thoughtful and insightful questions.
>
> **[Q1] Per-step latency and computational expense**
>
> **[A1]** We evaluated the inference latency per action chunk during simulation on an NVIDIA RTX 5090 GPU. The results are shown in the table below:
>
> |     | DP  | DP3 | GauDP |
> |-----|-----|-----|-------|
> | FPS | 1.49 | 1.57 | 1.28   |
>
> The inference latency of GauDP can be decomposed into two parts:
> - **Gaussian Estimation:** This component relies on a Transformer to compute self-attention and cross-attention. Thanks to extensive prior work on CUDA optimizations for Transformer computations, this part achieves high inference efficiency.
> - **Action Generation:** This part is identical to that of DP, except for a minor additional CNN module. Consequently, its latency is the same as DP, with the main latency arising from the iterative denoising process.
>
> Therefore, our GauDP method introduces only a marginal increase in inference latency compared to DP and DP3.
>
> While DP3 achieves the highest FPS in simulation, it's important to note that the point clouds used during DP3 simulation have been preprocessed to remove background and table clutter. This is feasible in simulation due to access to privileged semantics and optimized rendering. However, in real-world scenarios, acquiring high-quality point clouds and performing such preprocessing (e.g., filtering irrelevant points and applying farthest point sampling) is both time-consuming and computationally expensive. Therefore, simulation efficiency of point cloud-based policies may not translate directly to real-world deployments. In contrast, GauDP leverages 2D observations and reconstructs 3D Gaussians efficiently.
>
> **[Q2] Robustness to real-world visual disturbances and execution errors**
>
> **[A2]** Our Gaussian estimation module has been fine-tuned on large-scale datasets that already encompass diverse conditions, such as varying viewpoints and illumination. Additionally, we enhance the accuracy and robustness of Gaussian estimation by incorporating local depth constraints from multiple viewpoints. This design endows our method with inherent robustness against visual disturbances.
>
> To further evaluate robustness, we conducted experiments under novel settings that introduce challenging lighting variations—including random changes in light color, direction, and intensity—as well as random distractor objects (10 task-irrelevant objects randomly placed beside the manipulated objects on the table), following RoboTwin 2.0. In these settings, our method still achieved the highest task success rate:
>
> |             | DP  | DP3 | GauDP |
> |-------------|-----|-----|-------|
> | Grab Roller | 20% | 46% | 50%   |
>
>   **Robustness to execution errors**
>   - **Gaussian Estimation:** Regarding robustness to execution errors, our method reconstructs the 3D scene from two or more 2D observations. Since execution noise and actuation errors have limited impact on the 2D observations, they do not significantly affect the quality of the estimated Gaussians.
>   - **Action Generation:** The core contribution of our work is the proposal of a novel framework that leverages 3D Gaussian representations to coordinate 2D image inputs. Our policy learning technique remains consistent with that of DP and DP3, so the robustness of action generation to execution errors shares the same inherent limitations as diffusion policies. However, an interesting observation from our experiments is that policies based on GauDP tend to attempt re-execution of the current subtask more readily when encountering mid-task failures.
>
> **[Q3] Significant performance improvement on Lift Barrier task**
>
> **[A3]** Thank you for this insightful question. We believe that Lift Barrier and Grab Roller are particularly suitable for evaluating multi-agent coordination, as success requires tightly synchronized manipulation from both arms. Any premature or delayed movement by one agent typically results in failure.
>
> The Barrier object is made of thin white metal wires and appears as subtle lines in 2D observations, making it difficult to perceive using standard 2D encoders. In contrast, it is much more distinguishable in 3D point cloud representations, especially processed by FPS (Farthest Point Sampling), which explains the higher success rate of DP3 compared to DP.
>
> Our method bridges this gap by reconstructing 3D Gaussians from multiple 2D views in a self-supervised manner. This allows us to recover the spatial structure of challenging objects like the Barrier without requiring explicit 3D supervision or point cloud input. The improved success rate of GauDP on Lift Barrier demonstrates the effectiveness of our learned 3D representations for precise and coordinated manipulation.

---

> > ### Comment · Area_Chair_Hrqd · 2025-08-04
> >
> > Dear Reviewer,
> >
> > Please respond to the rebuttal. Thanks.
> >
> > AC.

---

> > ### Comment · Reviewer_o7bR · 2025-08-05
> >
> > The authors addressed my concerns properly. Admittedly, the method achieves higher success rates at the cost of a 2D Gaussian overhead. This is acceptable but not enough to qualify for a score of 5 in my opinion. Therefore, my score stays 4.

---

### Official Review · Reviewer_px8R · 2025-07-02

**Clarity:** 3
**Significance:** 3
**Originality:** 3
**Rating:** 5
**Confidence:** 4

**Summary:**

The paper studies the problem of multi-robot collaboration for manipulation tasks in the robotFactory benchmark. The paper introduces GauDP, a framework for training multi-agent coordination policies that integrate local and global observations using 3D Gaussian fields as a global representation for multi-agent embodied collaboration. The proposed method first reconstructs global 3D gaussian fields using each agents local view, next each agent dynamically queries shared gaussian representation to include additional information aggregated from multiple views. This is achieved by projecting the global 3D gaussian field back to agents original coordinate space. Next, the authors demonstrate the effectiveness and scalability of GauDP on the RoboFactory benchmark, achieving strong performance with only RGB input compared to other methods that use global point cloud representation and 2D image based policy learning.

**Questions:**

None

**Ethical Concerns:**

["NO or VERY MINOR ethics concerns only"]

**Limitations:**

The approach proposed in the paper is quite simple and effective what is lacking in the current paper is breadth of tasks in the experiments. This is understandable as the benchmarks for this setting do not have a lot of diverse tasks. I would recommend authors to add results for missing tasks from RoboFactory benchmark to make the current paper stronger given the benchmarks that exists already for completeness.

**Quality:**

3

**Strengths And Weaknesses:**

Strengths:

1. The paper is well-written and easy to follow
2. The approach proposed in the paper is novel, intuitive, simple and effective.
3. The proposed method outperforms other baselines that use only 2D images or builds global point cloud representations on majority of tasks from RoboFactory. The tasks where it underperforms 2D image based methods seems like those could be solved using a single agent more effectively and hence no improvement from effective collaboration.
4. The ablations in the paper are thorough and clearly show effectiveness of the global aggregation technique used by the authors. These experiments are quite insightful

Weaknesses:

1. It looks like the paper only includes results for a subset of tasks from RoboFactory. I believe the paper has a nice method that is effective so authors should include results on missing tasks in the main paper (ex: pass shoe in 2 agent setup, long pipeline delivery in 4 agent).
2. It is unclear why the task stack cube has such low or 0 success rates. Additional analysis highlighting failure modes would help build insights about what is missing in the current methods. It seems like these tasks should be solvable by a single agent itself so it seems counterintuitive that it is not working in multi-agent setting.

---

> ### Author Rebuttal · Authors · 2025-07-31
>
> **[W1] missing tasks of RoboFactory in the main paper**
>
> **[A1]** Thank you for your kind recognition and valuable suggestion. We agree that including more diverse tasks is important for comprehensive evaluation. However, the tasks "pass shoe (2-agent)" and "long pipeline delivery (4-agent)" are relatively weakly cooperative in nature. These tasks can be decomposed into sequential pick-and-place subtasks that each agent can complete independently with minimal coordination. Since our framework is designed to emphasize and promote **tight and synergistic collaboration** among agents, we focused on tasks that require strong inter-agent dependency and coordinated decision-making.
>
> **[W2] "Stack cubes" tasks have a low or 0 success rate**
>
> **[A2]** We appreciate your insightful comment. Indeed, the “stack cubes” tasks are challenging despite appearing simple at first glance. There are two major contributing factors to their low success rates:
> - **Accumulated manipulation difficulty:** Stacking cubes requires precise pick-and-place actions for each individual cube. Even a small deviation in position can cause failure in stacking. Given that several such precise actions must be executed in sequence, the overall success rate decreases approximately exponentially with the number of cubes.
> - **High task complexity due to spatial and sequential constraints:** Each cube has a unique color and must be stacked in a specific order. Agents must select the appropriate cube based on color and determine which agent is closest and most suitable to pick it up. Since cube positions are randomly initialized, the spatial layout and action sequence vary significantly across demonstrations. This introduces substantial variability and demands that the policy generalize across many permutations—much more than in other tasks.
>
> To further investigate, we fine-tuned Pi0 (a strong VLA-based method) on the “stack cube three” task. The success rate remained extremely low (~1%), suggesting that the difficulty arises not from our specific method, but from the fundamental complexity of the task.
>
> In summary, the “stack cubes” tasks uniquely combine the need for high-precision manipulation, spatial reasoning, and decision-making under dynamic layouts. These factors make them the most difficult among our experiments, and we hope our analysis helps clarify the observed performance gap.

---

> ### Comment · Area_Chair_Hrqd · 2025-08-07
>
> Dear Reviewer,
>
> Please comment on the rebuttal ASAP. Thanks.
>
> AC.

---

### Official Review · Reviewer_zMRB · 2025-07-08

**Clarity:** 3
**Significance:** 3
**Originality:** 3
**Rating:** 5
**Confidence:** 4

**Summary:**

The paper introduces GauDP, where a 3D gaussian splat representation generated using a (fine-tuned) MLP is used in conjunction with a ViT to generate a unified representation of the scene that is conducive to multi-agent robotics scenarios. They show that their method performs better than previous baselines in a simulated multi-robot environment, and explain some failure modes that other sensing methods such as local image observations and point clouds exhibit.

**Questions:**

What are the exact setup parameters for your diffusion policy (and baselines)? I would like to see all relevant hyperparameters such as action prediction horizon, execution horizon, scheduler parameters, etc.
How much time does your method require to train and infer to recreate the results in the paper, and how does this compare to your baselines?
The final version of this paper should have details and background on diffusion policy in the preliminaries. I believe you can shorten the ablations section significantly while conveying the same information to fit this in.
I would like to see a search over relevant hyperparameters, such as the action (prediction, execution) horizon, and possibly normalize based on training flops if training times are long. This could be very beneficial to your paper: intuitively, I can imagine that GauDP would scale well with longer prediction horizons due to increased precision and coordination. I disagree with the notion that identical hyperparameters allow for a fair comparison (especially since the architectures used seem to be different). In fact, it would be the exact opposite as there is a chance the hyperparameters were tuned to your method (e.g., long action horizon), and these choices might have been incompatible with others. What if using half the action horizon reduced the occurrences of collision without the need for the 3D representation? It is possible that some errors would not happen if given the time to react (perhaps even using a smaller execution horizon might show this).
How many parameters is your ViT and splatting MLP (combined)? Additionally, how many parameters are in each baseline? I see in Sec. 4.1 that the 2D baselines were ResNet 18, this is quite a small model. If you were to flop or parameter-normalize each method, how do the results change?

If these questions are addressed (and the relative performance of GauDP holds), I would be inclined to increase my score as the results seem otherwise strong and intuitive.

**Ethical Concerns:**

["NO or VERY MINOR ethics concerns only"]

**Final Justification:**

The authors have added crucial training details (majority of the reason for the original rating), and added experiments regarding model size normalization. All of the results check out, and the downsides are now clear (lower throughput and longer training time). However, the downsides are mitigated by stronger empirical performance.

**Limitations:**

See above

**Quality:**

3

**Strengths And Weaknesses:**

Strengths:
Methodology is simple and achieves strong results in the domain they test. Substantially outperforms baselines.
Method is explained clearly, and it is intuitive why performance should be better than the baselines.
Ablations show that parts of the pipeline are not redundant.

Weaknesses:
In general, the paper is devoid of any important hyperparameters required to reproduce the material. For instance, it is completely missing any details on their diffusion policy setup, such as action prediction horizon, action execution horizon, denoising steps/schedule, etc., none of these details are in the appendix. This should be added to the main paper for clarity.
There is no information about runtimes of different methods, training times/flop estimates, inference times, etc. I would like to see information about the throughput of each system. For example, how much slower is GauDP (if it is, this method seems computationally expensive) compared to the point cloud and local context only methods?
In section 4.1, they claim that they use identical hyperparameters for “fair comparison”. I disagree with this, and explain why in question 3.
Comparisons are potentially unfair, 2D representations seem to be using a smaller model, though details on exact model sizes or parameters for GauDP are not provided in either the supplementary or main paper.

---

> ### Author Rebuttal · Authors · 2025-07-31
>
> We sincerely thank the reviewer for the detailed and constructive feedback. We appreciate the suggestions regarding reproducibility, fairness of comparison, and computational analysis. **All of our hyperparameters were aligned with the configurations used in prior work**. We will include additional hyperparameter details in the main paper and release all related code and configuration files to ensure full reproducibility. We have carefully addressed each point as follows:
>
> **[A1] Hyperparameter Details for Reproducibility**
>
> We agree with the reviewer that providing complete hyperparameter settings is essential for reproducibility. We will include these details in the revised version of the main paper. Specifically, our proposed GauDP and all baseline methods adopt identical hyperparameters following prior diffusion policy works and existing benchmarks:
> - Action prediction horizon: 8
> - Number of observation steps: 3
> - Number of action execution steps: 6
> - Denoising schedule: Our experiments follow the optimal configurations of DP and DP3 to ensure fair comparison. Specifically, both GauDP and DP adopt DDPM with 100 denoising steps for both training and inference, while DP3 uses DDIM, also with 100 steps.
>
> **[A2] Training/Inference Time, Parameter Count**
>
> We have conducted a thorough analysis of training and inference runtimes, as well as model size.
>
> Training time depends on the trajectory length of each task.  Take the Lift Barrier task as an example, the number of elapsed steps is approximately 100:
>
> - **Training time** (100 epochs on A100 GPU):
>   - DP: 4.8 GPU hours
>   - DP3: 2.5 GPU hours
>   - GauDP: 6.5 GPU hours
>
>   **GauDP has comparable training time to DP**, thanks to two factors:
>   - The Gaussian estimation module is **frozen** during the whole policy training phase;
>   - We implemented extensive optimizations on data I/O and training efficiency for all these methods. (The relevant codes will be released after the review process.)
> - **Inference speed of action chunk** (Simulation evaluation on NVIDIA 5090 GPU):
>   - DP: 1.49 FPS
>   - DP3: 1.57 FPS
>   - GauDP: 1.28 FPS
> - **Parameter counts** (depend on the number of agents):
>
> *Table 1*
>   |         | DP  | **GauDP-policy** | LargeDP | GauDP-full |
>   |:-----:|:-----:|:-----:|:-----:|:-----:|
>   | 2 Agents | 129.949M | 129.959M | 721.286M |750.505M |
>   | 3 Agents | 163.584M |  163.594M |  956.335M | 784.140M |
>   | 4 Agents | 197.130M | 197.140M | 1.191G | 817.686M |
>
> In the table above, **GauDP-full** refers to the complete model including the Gaussian estimation module, while **GauDP-policy** only includes the policy learning component. To ensure a fair comparison in terms of parameter count, we introduce **LargeDP**, which scales up the original DP model to match the parameter size of *GauDP-full*. The corresponding performance is reported in *Table 2*. It is also important to note that the majority of parameters in *GauDP* are frozen during policy learning, whereas all parameters in *LargeDP* are learnable. This comparison highlights that *GauDP* introduces minimal overhead within the policy module, yet achieves a significant improvement in success rate, demonstrating the effectiveness of our framework in enhancing multi-arm coordination.
>
> **[A3] Fairness of Comparison**
>
> We appreciate the reviewer’s concern regarding fair comparison. To clarify, our policy architecture is **strictly aligned with DP**: we **use the same observation encoder (Resnet18) and the same 1D conditional diffusion module as DP**. The only additions in GauDP are:
> - A fine-tuned NoPoSplat module with local depth constraints for estimating the Gaussian representation;
> - Two 3-layer CNNs: one for channel adjustment and one for pre-fusion of the Gaussian image.
>
> This design choice allows us to isolate and evaluate the benefits of our proposed **Gaussian-Image Synergy framework** under controlled conditions. By keeping the core policy module unchanged, we ensure that observed improvements result from the proposed representation rather than from architectural scaling. We will further clarify the model architecture details in the supplementary material to avoid any potential confusion.
>
> **[A4] Model Size Normalization**
>
> We further scaled up the baseline DP model into **LargeDP** to match or even exceed the parameter count of GauDP, as shown in *Table 1*. GauDP still achieves the highest success rate, as shown in *Table 2*, reinforcing that the performance gain stems from our proposed framework rather than increased model capacity. LargeDP still lacks an effective representation for global perception and fine-grained understanding of 3D structure and spatial relationships. This further supports the effectiveness of our method in facilitating multi-agent cooperation.
>
> *Table 2*
> | Model   | Lift Barrier | Place Food | Stack 2 Cubes | Align Camera | Stack 3 Cubes | Take Photo | Average |
> |:-----:|:-----:|:-----:|:-----:|:-----:|:-----:|:-----:|:-----:|
> | DP      | 9            | 12         | 6              | 3             | 0              | 0           | 5       |
> | DP3     | 30           | 21         | 1              | 3             | 0              | 9           | 10.67   |
> | LargeDP | 60           | 12         | 4              | 29            | 0              | 0           | 17.5    |
> | **GauDP**   | 72           | 15         | 2              | 26            | 0              | 3           | **19.67**   |
>
> **[A5] Revisions to Main Text and Supplementary Materials**
>
> We will include additional hyperparameter details in the main paper and provide background information on diffusion policy in the preliminaries. Additionally, we will shorten the ablation study section in the main paper and provide more detailed descriptions of the model architecture in the supplementary materials accordingly in the revised version.

---

> > ### Comment · Area_Chair_Hrqd · 2025-08-04
> >
> > Dear Reviewer,
> >
> > Please respond to the rebuttal. Thanks.
> >
> > AC.

---

> > ### Comment · Reviewer_zMRB · 2025-08-04
> >
> > I thank the authors for their analysis and explicit clarifications on training setup. While it seems like GauDP does indeed have computational overhead over other methods, I believe that this is acceptable due to the increase in performance (though it seems normalizing model sizes has made the results closer). I have increased my score accordingly to a 5 as they have addressed my concerns.

---

### Decision · Program_Chairs · 2025-09-17

**Decision:**

Accept (poster)

**Comment:**

This paper presents GauDP, a framework that uses 3D Gaussian field representations reconstructed from multiple agents' RGB observations to enable effective multi-agent robotic collaboration. The technical approach bridges 2D image-based and 3D point cloud methods by allowing agents to query a shared 3D scene representation for enhanced spatial understanding during manipulation tasks. Initial reviewer concerns regarding reproducibility, computational efficiency, and experimental fairness were comprehensively addressed through detailed rebuttals that provided complete hyperparameters, thorough computational analysis showing acceptable overhead, and fair model size comparisons. The experimental validation on RoboFactory demonstrates clear performance improvements over baselines. The work makes a solid technical contribution to multi-agent robotics with practical value, and all reviewers ultimately recommended acceptance after their concerns were adequately resolved.